# Effect of Grain Refiner on Fracture Toughness of 7050 Ingot and Plate

**DOI:** 10.3390/ma14216705

**Published:** 2021-11-07

**Authors:** Fang Yu, Xiangjie Wang, Tongjian Huang, Daiyi Chao

**Affiliations:** 1School of Material Science and Engineering, Northeastern University, Shenyang 110044, China; wangxj@epm.neu.edu.cn; 2Shandong Nanshan Aluminum Co., Ltd., Longkou 265713, China; htj20141119@icloud.com (T.H.); cdy19861226@163.com (D.C.)

**Keywords:** Al-5Ti-0.2B, Al-3Ti-0.15C, 7050 ingot, 7050-T7651 plate, fracture toughness, grain size

## Abstract

In this paper, two types of grain refining alloys, Al-3Ti-0.15C and Al-5Ti-0.2B, were used to cast two types of 7050 rolling ingots. The effect of Al-3Ti-0.15C and Al-5Ti-0.2B grain refiners on fracture toughness in different directions for 7050 ingots after heat treatment and 7050-T7651 plates was investigated using optical electron microscopy (OEM) and scanning electron microscopy (SEM). Mechanical properties testing included both tensile and plane strain fracture toughness (K_IC_). The grain size was measured from the surface to the center of the 7050 ingots with two different grain refiners. The fracture surface was analyzed by SEM and energy dispersive spectrometer (EDS). The experiments showed the grain size from edge to center was reduced in 7050 ingots with both the TiC and TiB refiners, and the grain size was larger for ingots with the Al-3Ti-0.15C grain refiner at the same position. The tensile properties of 7050 ingots after heat treatment with Al-3Ti-0.15C grain refiner were 1–2 MPa lower than the ingot with the Al-5Ti-0.2B grain refiner. For the 7050-T7651 100 mm thick plate with the Al-3Ti-0.15C grain refiner, for the L direction, the tensile properties were lower by about 10~15 MPa; for the plate with the Al-3Ti-0.15C refiner than plate with Al-5Ti-0.2B refiner, for the LT direction, the tensile properties were lower by about 13–18 MPa; and for the ST direction, they were lower by about 8–10 MPa compared to that of Al-5Ti-0.2B refiner. The fracture toughness of the 7050-T7651 plate produced using the Al-3Ti-0.15C ingot was approximately 2–6 MPa · $\sqrt{m}$ higher than the plate produced from the Al-5Ti-0.2B ingot. Fractography of the failed fracture toughness specimens revealed that the path of crack propagation of the 7050 ingot after heat treatment produced from the Al-3Ti-0.15C grain refiner was more tortuous than in the ingot produced from the Al-5Ti-0.2B, which resulted in higher fracture toughness.

## 1. Introduction

7050 Aluminum alloy have been widely used in the aerospace industry due to their low quench sensitivity, high fracture toughness, high fatigue crack growth rate (FCGR), and excellent stress corrosion resistance [1]. With the development of the aviation industry, the requirement for comprehensive performance of materials is getting higher, which means the raw material needs to have higher fracture toughness, higher FCGR, and high fatigue properties [2,3,4]. Normally, the tensile property levels decrease as the toughness level increases by alloying and heat treatment [5]. Therefore, the goal that most aluminum manufacturers want to pursue is to improve the fracture toughness of 7050 plates without the loss of the tensile property.

The effect of the microstructure on aluminum fracture toughness has been studied by many researchers. The main factors affecting fracture toughness [5,6] are grain size, recrystallization ratio, coarse insoluble phase, and strengthening precipitates. The relationship between recrystallization fraction (large angle grain boundary volume) and fracture toughness [7,8] was quantitatively analyzed. The higher the recrystallization fraction, the lower the fracture toughness. Hahn et al. [5] found that reducing the Fe, Si content and increasing the amount of smaller typical Cr, Mn, Zr-bearing particles could improve facture toughness, and the ratio of Zn:Mg [9] could also influence the fracture toughness. Zhang, Dumont et al. [10,11,12] investigated the effect of heat solution treatment process and quench parameters on the fracture toughness of 7050 and 7085 thick plate. Chen et al. [13] observed that the deformation texture can improve the fracture toughness and tensile properties of 7B50-T7751 plate effectively, and increase different directions of anisotropy. Other researchers also reported the effect of homogenization process [14] and aging parameter [15] on fracture toughness.

Although many experiments have been conducted to observe the factors that affect fracture toughness, there are still certain issues that have not been clarified. The grain size is known to be one of the major factors that affects fracture toughness [5], and grain refiners decide the final grain size and the quality of the ingot that is used for the manufacture of aviation products. A large number of studies have focused on the effect of grain refiners on the grain size and mechanical properties of ingots or products [16,17,18,19,20,21]. The effect of Al-5Ti-0.25C and Al-5Ti-0.2B additions on the grain size of different wrought alloys (AA1050, AA3004, AA5182, AA6063, AA7050, AA7475, AA8079) under different melting temperatures and addition rates was investigated [16] and it was found that the grain size decreased when increasing the rate of the refiner, and in all alloys, with the exception of AA3004, fine grain sizes (130 µm) were achieved with the addition of 1 kg/t. Nagaumi et al. [17] studied the effect of Al-3Ti-0.2B and Al-5Ti-1B on the microstructure of 7050 ingots and found that the refining fading of the Al-3Ti-0.2C refiner was more evident than that of Al-5Ti-1B grain refiner when the soaking time was increased. Huang et al. [18] noticed that compared with the 7050 alloy using the Al-5Ti-1B grain refiner, the distribution of the second phases present in 7050 aluminum alloy was more dispersed and uniform than the alloy using Al-5Ti-0.2C refiner. They also found that increasing the amount of Al-5Ti-0.2C grain refiner was more effective at easing the “Zr poisoning” phenomenon of 7050 alloy and could improve the strength and hardness while keeping good elongation. There are few reports [22] that discuss the relationship between grain refiner and fracture toughness of aviation aluminum products.

To examine the relationship between grain refiner and fracture toughness, a systematic study was conducted in 7050 ingots and 7050-T7651 100 mm thick plate with the grain refiner of Al-3Ti-0.15C and Al-5Ti-0.2B, respectively. The prepared plate was subjected to the same heat solution treatment and aging process. Microstructure characterization was then performed to evaluate the difference of the 7050 ingot and plate with two different grain refiners, and this study has guiding significance for industrial production of the 7050-T7651 plate, which needs higher fracture toughness.

## 2. Materials and Methods

The materials used for this study were commercial 7050 ingots with Al-3Ti-0.15C and Al-5Ti-0.2B grain refiner provided by Shan Dong Nanshan Aluminum Co., Ltd. at Longkou, China. The feeding amount for both Al-3Ti-0.15C and Al-5Ti-0.2B grain refiner was 2.5 kg/t. The ingot size was 440 mm × 1420 mm × 5000 mm and the composition is shown in Table 1.

After casting was completed, slices of the two different grain refined ingots were taken to evaluate the microstructure, grain size, and mechanical properties of the as-cast material according to the sampling diagram in Figure 1. The ingot samples were solution heat treated and artificially aged (heat solution treatment and age process were 479 °C × 50 min and 121 °C × 4 h + 163 °C × 20 h, respectively) prior to mechanical testing to eliminate the effects of the hot rolling process. The tensile properties and fracture toughness of ingot after heat treatment were then measured by an Instron 5985 tensile machine (INSTRON, Boston, MA, USA) and Instron 8802fatigue test machine (INSTRON, Boston, MA, USA), respectively. The fracture surface for ingots with different grain refiners were analyzed by scanning electron microscopy (SEM, FEI, Hillsboro, OR, USA).

A 100 mm plate for the 7050 alloy was hot rolled from ingots that were cast with two different types of grain refining alloy. The first set of 7050 plates were produced using Al-3Ti-0.15C grain refined ingots, and the second set of 7050 plates were produced using Al-5Ti-0.2B grain refined ingots according to the preparation process shown in Figure 2. Ensuring the consistency of thermomechanical processing enables the impact of grain refining on the wrought microstructure and mechanical properties to be isolated. All of the TiC and TiB_2_ AA7050 ingots were homogenized together in the same furnace using a two-step process (step I: 465 °C soaking for 8 h, step II: 478 °C soaking for 20 h). After the homogenization process was finished, the ingots with Al-3Ti-0.15C and Al-5Ti-0.2B refiner were hot rolled to a 100 mm plate, then solution heat treated and artificially aged together to ensure consistent thermomechanical processes and to get the 7050 plate with T7651 temper (7050-T7651 plate).The heat solution heat treatment process was soaking 240 min at 479 °C, and the aging was a two-step process: Step I was soaking 4 h at 121 °C and Step II was holding 18 h at 163 °C.

Samples for the 100 mm 7050-T7651 plate were taken according to the requirement of the AMS2355 standard for tensile property and fracture toughness test, and the tensile properties and fracture toughness were tested on T/4 of plate by an Instron 5985 tensile machine and Instron 8802 fatigue test machine, respectively. Specimen orientations for fracture toughness are shown in Figure 3.

## 3. Results

### 3.1. The Effect of Grain Refiner on 7050 Ingot Grain Size

The grain size from the edge to the center of the 7050 ingot along with the casting direction was observed, five pictures were selected for each position, and the average grain size was analyzed by the intersection method according to ASTM E112 standard [23]. The microstructure and calculated grain size of the ingot with Al-5Ti-0.2B and Al-3Ti-0.15C grain refiner are shown in Figure 4 and Figure 5, respectively. From Figure 4, the grain size from edge to center is increased, with a grain size of 95.99 μm, 101.04 μm, and 121.96 μm from edge to center for the ingot with Al-5Ti-0.2B grain refiner and 127.31 μm, 140.50 μm, and 165.32 μm with Al-3Ti-0.15C grain refiner, respectively. According to the study [17] by Hiromi, the average grain size of the 7050 ingot without grain refiner was 210 μm, and both the Al-5Ti-0.2B and Al-3Ti-0.15C grain refiner have refined the grain size.

### 3.2. The Effect of Grain Refiner on Tensile Property of 7050 Ingot and 7050-T7651 100 mm Plate

Table 2 shows the tensile properties of the 7050 ingot along with casting direction after heat treatment with Al-5Ti-0.2B and Al-3Ti-0.15C grain refiners. The tensile property of the 7050 ingot with Al-3Ti-0.15C grain refiner is a little lower than that of the ingot with Al-5Ti-0.2B grain refiner.

Table 3 depicts the tensile properties of 7050-T7651 100 mm thick plate with different grain refiners. According to the requirements of AMS 2355 standard [24], the mechanical properties at 1/4 thickness should be tested for plates with thickness of more than 38.1 mm. Therefore, for 7050-T7651 100 mm thick plate, the mechanical properties were tested at the location of 1/4 thickness to investigate the influence of different grain refiners on the mechanical properties. As can be seen from Table 3, compared with the addition of Al-5Ti-0.2B refiner, the tensile properties of 7050-T7651 100 mm thick plate with the addition of Al-3Ti-0.15C grain refiner in all directions are slightly reduced, and the elongation is not much different. For the L direction, the tensile property is lower by about 10–15 MPa for the plate with Al-3Ti-0.15C refiner than for the plate with Al-5Ti-0.2B refiner; for the LT direction, the tensile property is lower by about 13–18 MPa than that of Al-5Ti-0.2B refiner; and for the ST direction, the yield strength is lower by about 8–10 MPa than that of Al-5Ti-0.2B refiner.

### 3.3. Effect of Different Grain Refiners on Fracture Toughness of 7050 Ingot and Plate

Figure 6 shows fracture toughness test results for the 7050 ingot along with casting direction and 7050-T7651 100 mm plate at 1/4 thickness with different grain refiners. The fracture toughness of the ingot is increased by the addition of Al-3Ti-0.15C grain refiner. Comparing fracture toughness for different directions of the 7050-T7651 100 mm plate with Al-5Ti-0.2B grain refiner, plates with Al-3Ti-0.15C grain refiner have higher fracture toughness at L-T, T-L, and S-L directions, and the difference of the fracture toughness for the L-T direction for the 7050 product with different grain refiners is more obvious. The fracture toughness of the 7050-T7651 plate with Al-3Ti-0.15C grain refiner for the L-T direction is about 5 MPa · $\sqrt{m}$ higher than for the plate with Al-5Ti-0.2B grain refiner.

The fracture surfaces of the 7050 ingot fracture toughness specimens after heat treatment were analyzed by SEM to observe the effect of the grain refiners. Figure 7 shows fractography for different grain refiners of the ingots after heat treatment. The fracture surface exhibits different character for ingots with different grain refiners. Intergranular failure is observed linking shear planes from adjacent grains in ingots with Al-5Ti-0.2B grain refiner, whereas ingots with Al-3Ti-0.15C grain refiner exhibits transgranular failure via sheared planes covered with fine dimples. The crack path of the ingot with Al-3Ti-0.15C grain refiner is more tortuous.

The EBSD analysis results of different refiners for ingots after heat treatment at the crack tip are shown in Figure 8. As can be seen from Figure 8, the ingot containing Al-3Ti-0.15C refiner has intergranular and transgranular fracture, and the ingot produced by Al-5Ti-0.2B refiner presents intergranular fracture. Moreover, the ingot microstructure has a genetic effect: the grain size of the ingot with Al-3Ti-0.15C grain refiner is larger under the same refiner adding amount and casting process compared with Al-5Ti-0.2B grain refiner, which leads to increase in fracture toughness of the final product.

SEM analysis of the fracture surface for 7050-T7651 100 mm thick plate with different refiners were carried out. The samples in the T-L direction have both intergranular and transgranular fractures, and a large amount of second phases are distributed at the same time. The enlarged SEM photos of the fracture surfaces with different grain refiners are shown in Figure 9. The energy dispersive spectrometer (EDS) analysis shows both TiB_2_ and TiC particles have a granular shape, and the size of TiB_2_ particles is larger than that of TiC particles; the TiC particles were difficult to find under the scope of SEM. In addition, TiB_2_ particles are more aggregated than TiC particles, whereas TiC particles are more dispersed and have stronger resistance to crack propagation. The basic characteristics of fracture are basically consistent with the paper [25] researched by Zhang Xinming from Central South University.

## 4. Discussion

The ingot is solidified from the surface to the center in turn. The surface layer has a high degree of supercooling, high nucleation rate, and relatively small grains because of the faster heat dissipation. Moreover, heat dissipation of the center is slow, the degree of undercooling is small, and the nucleation rate is low, resulting in larger grains in the center. Literature [18] shows that the refining effect of Al-5Ti-0.2B is better than that of Al-3Ti-0.15C, primarily because of the microstructure of the refiner and the morphology and distribution of the refined particles. As shown in Figure 10, TiB_2_ crystal in the Al-5Ti-0.2B grain refiner has a closed hexagonal structure, which cannot provide a nucleation core for the matrix. The thin Al_3_Ti layer is enriched on the {0001} plane, which is used as the basis to promote α-Al nucleation. The nucleation directions of Al_3_Ti are <110>{112} and <210>{112}, respectively. During solidification, Al_3_Ti can nucleate in multiple directions and form a “halo” structure with “circle” shape; TiC crystal has face centered cubic (FCC) structure, which can be used as grain refining core for nucleation at the {111} plane and form “petalous” shape, and the number of nucleation is lower than that of the Al-Ti-B refiner. It is also related to the microstructure of the grain refiner and the morphology and distribution of the particles. The literature [17] shows that TiC particles in Al-3Ti-0.15C are unstable. When held in a soaking furnace for a long time, TiC particles will aggregate and react to form more stable Al_4_C_3_, whose refining effect is far less than that of TiC particles. Although for Al-Ti-B grain refiner, long soaking time also causes agglomeration and deposition of TiB_2_ particles, studies have shown [21] that Al-Ti-B is still effective in refining pure aluminum, even if the holding time in the furnace is as long as 24 h. Combined with the above factors, the grain refinement effect of Al-3Ti-0.15C is not as good as that of Al-5Ti-0.2B. Therefore, with the same amount of addition, the grain size of the ingot with Al-5Ti-0.2B grain refiner is smaller.

If the grain size is small enough, plasticity will be enhanced without detrimental, low energy, intergranular fracture for thin products under plane stress. However, for thick products under plane strain, fracture is usually controlled by coarse particles, and a recrystallized grain structure is preferable. Teleshov [26] studied the effect of grain size on fracture toughness and strength of AK4-1CH extrusion products. It was shown that the larger grain size products have higher fracture toughness than the smaller grain size products in L-T, T-L, and S-L directions, especially in the T-L direction. By comparing the microstructure, properties, and the distance between the vertical crack plane and the second phase in the propagation direction, it was found that the main reason for the different fracture toughness of AK4-1CH products may be the smaller extent of grain boundary in the path of plastic deformation. To overcome this deformation, the deformation in neighboring grains is needed. However, the stress generated far from the crack tip is not enough, which leads to the increase in the size of the plastic deformation zone, resulting in the increase in KIC. According to the study [27] of Krasovskii, the depth of the plastic deformation zone is related to the fracture toughness and yield strength of the product. According to the equation λ = 0.128 ∗ (KIC/δ_0.2_)^2^, the value for the ingot with different grain refiners is 1.080 mm and 0.787 mm, respectively, which are larger than the average grain size. Fracture path for the 7050 ingot with different grain refiners is shown in Figure 11. T range of the plastic deformation zone with large grain size was large, which affected the fracture mode at the crack tip.

As discussed in Section 2, the recrystallization ratio is one of the factors affecting fracture toughness: the higher the recrystallization fraction, the lower the fracture toughness. The recrystallization ratio of the 7050-T7651 plate in different thickness positions with Al-5Ti-0.2B and Al-3Ti-0.15C grain refiners are shown in Figure 12. From surface to 1/2 thickness, the recrystallization ratio for the 7050 plate with Al-5Ti-0.2B grain refiner is 11.2%, 10.3%, and 2.4%, respectively, and 12.3%, 8.1%, and 1.5% with Al-5Ti-0.2B grain refiner. There are no obvious differences in the recrystallization ratio for the 7050-T7651 plate with different grain refiners.

The Fe and Si bearing phase formed during solidification is brittle and not coherent with the matrix, easily forms pores, and becomes a source of crack propagation, thus strongly reducing the fracture toughness of the alloy. The size of the precipitate and the interface binding force between the precipitate and the matrix have a great influence on the fracture toughness. The larger the size of the precipitate, the weaker the binding force between the phase and the matrix, which leads to the crack occurrence and the inferior fracture toughness. Gokhale, Deshpande et al. [7,28] quantitatively analyzed the relationship between microstructure and fracture toughness, and found that the larger the surface area of the second phase particle per unit volume, the lower the fracture toughness. Therefore, in order to improve the fracture toughness of 7050 products, the content of Fe and Si and the size of the dispersion phase must be strictly controlled. In the 7050-T7651 plate with Al-5Ti-0.2B grain refiner, TiB_2_ particles are easy to gather in the process of casting and have a genetic effect; TiB_2_ particles act as heterogeneous nucleation to precipitate in the aging process. The large precipitate phase will be become the source of cracks, which may reduce the local plastic deformation ability and also the fracture toughness.

## 5. Conclusions

A study has been performed on the effect of grain refiner on the tensile property and fracture toughness of 7050 ingots and 7050-T7651 100 mm thick plate. The following conclusions may be drawn from this work.

The experiments showed that the grain size from surface to center of the ingot with the addition of Al-5Ti-0.2B grain refiner was 95.99 μm,101.04 μm, and 121.96 μm, respectively, and 127.31 μm, 140.50 µm, and 165.32 µm with the addition of Al-3Ti-0.15C grain refiner. The refining ability of Al-5Ti-0.2B grain refiner was better than that of Al-3Ti-0.15C.

For both 7050 ingots and 7050-T7651 100 mm thick plate, the tensile property with Al-5Ti-0.2B grain refiner was higher than that of Al-3Ti-0.15C grain refiner. However, the fracture toughness of 7050 ingots and 7050-T7651 plates made with Al-3Ti-0.15C grain refiner was higher than that made with Al-5Ti-0.2B grain refiner, and fracture path for the 7050 ingot with Al-3Ti-0.15C after heat treatment was more tortuous than for the ingot with Al-5Ti-0.2B.

The fractured surface for the 7050-T7651 plate showed both TiB_2_ and TiC particles have a granular shape, and the size of the TiB_2_ particles was larger than that of the TiC particles. TiC particles were hard to find under the scope of SEM, whereas TiB_2_ particles were easy to find and easy to aggregate. The combination of grain size and uniform distribution of the second phase results in higher fracture toughness for 7050 ingots and 7050-T7651 plate with Al-3Ti-0.15C grain refiner.

## Figures and Tables

**Figure 1 materials-14-06705-f001:**
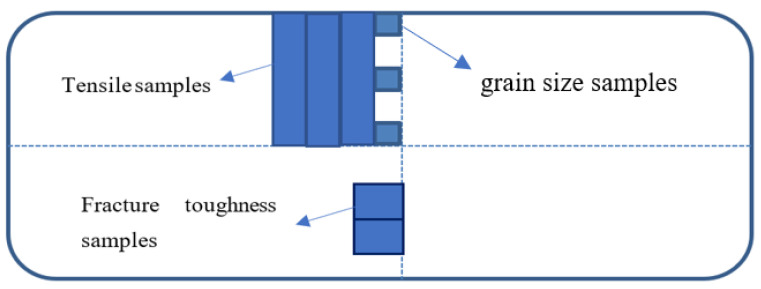
Samples locations for 7050 ingot.

**Figure 2 materials-14-06705-f002:**
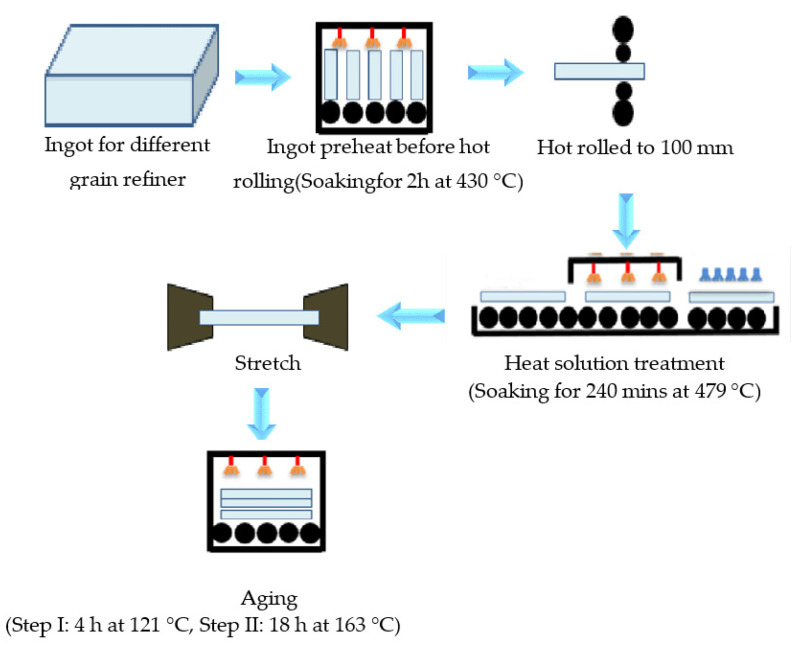
Scheme of the preparation process of the studied 7050-T7651 plate.

**Figure 3 materials-14-06705-f003:**
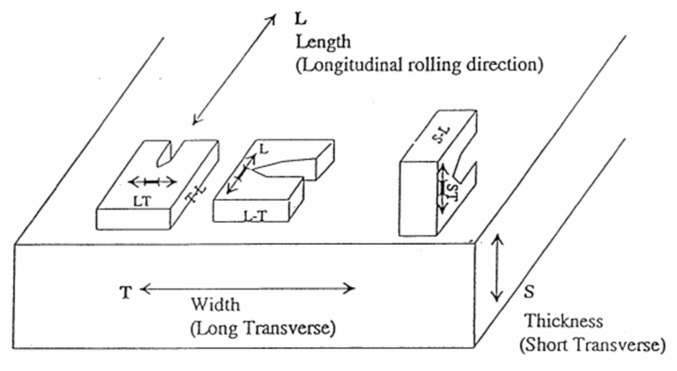
Specimen orientations used for fracture toughness test.

**Figure 4 materials-14-06705-f004:**
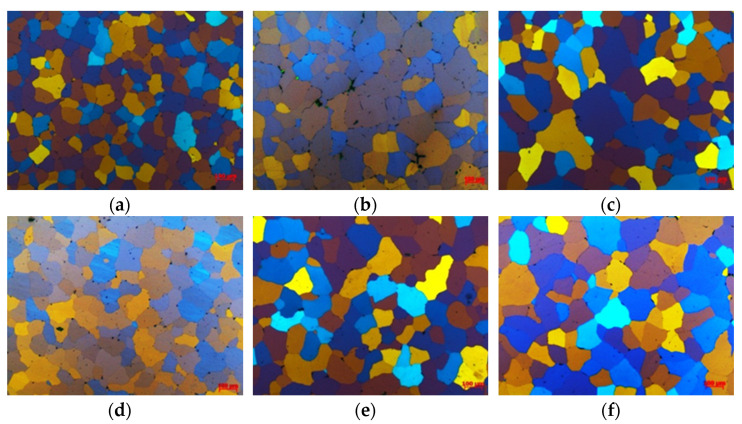
Grain structure of 7050 ingot: (**a**) Al-5Ti-0.2B surface, (**b**) Al-5Ti-0.2B 1/4 thickness, (**c**) Al-5Ti-0.2B 1/2 thickness, (**d**) Al-3Ti-0.15C surface, (**e**) Al-3Ti-0.15C 1/4 thickness, and (**f**) Al-3Ti-0.15C 1/2 thickness.

**Figure 5 materials-14-06705-f005:**
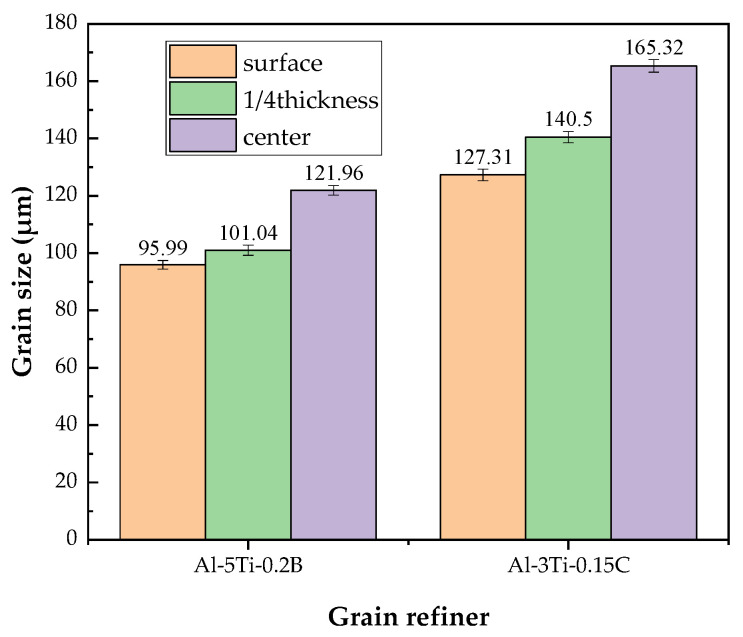
Average grain size for ingot using different grain refiners.

**Figure 6 materials-14-06705-f006:**
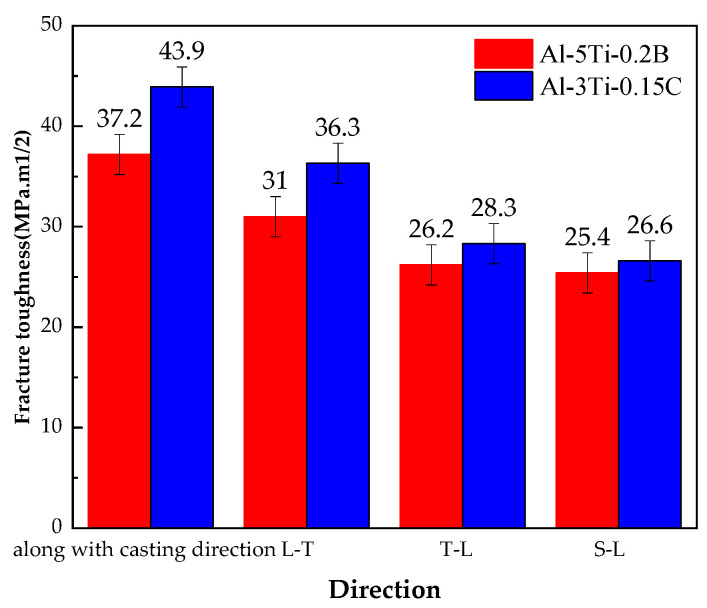
Fracture toughness of 7050 ingot after heat treatment and 7050-T7651 100 mm plate for different grain refiners in different directions.

**Figure 7 materials-14-06705-f007:**
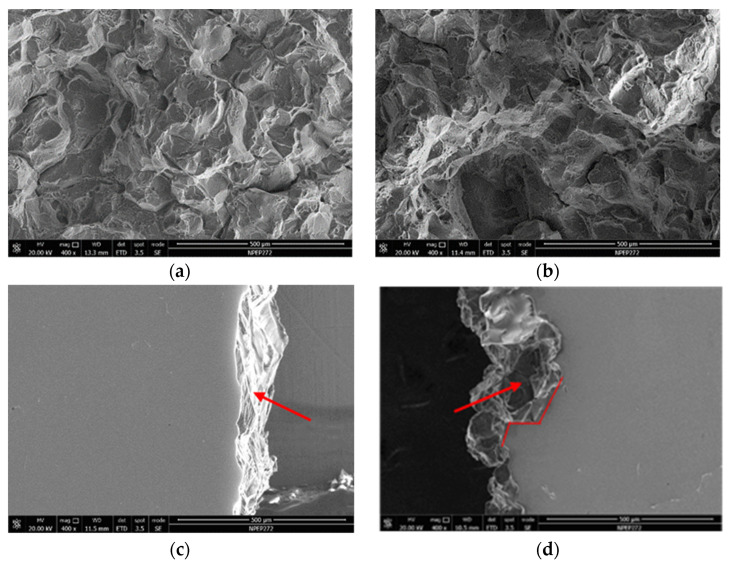
Fracture surfaces of ingot for different grain refiners: (**a**,**c**) Al-5Ti-0.2B grain refiner, (**b**,**d**) Al-3Ti-0.15C grain refiner.

**Figure 8 materials-14-06705-f008:**
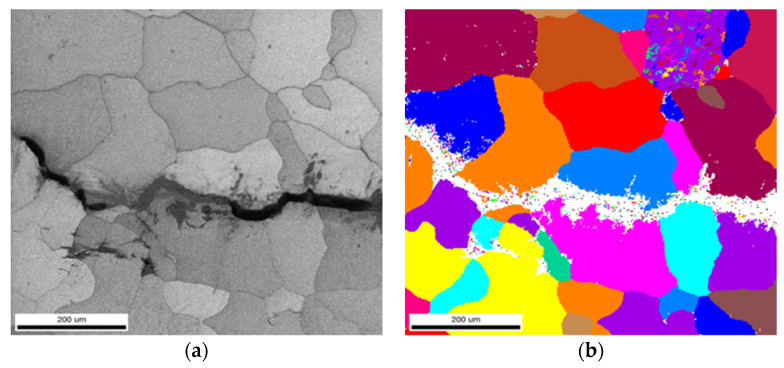
EBSD analysis for 7050 ingot fracture surface at crack tip: (**a**,**b**) ingot with Al-3Ti-0.15C grain refiner, (**c**,**d**) ingot with Al-5Ti-0.2B grain refiner.

**Figure 9 materials-14-06705-f009:**
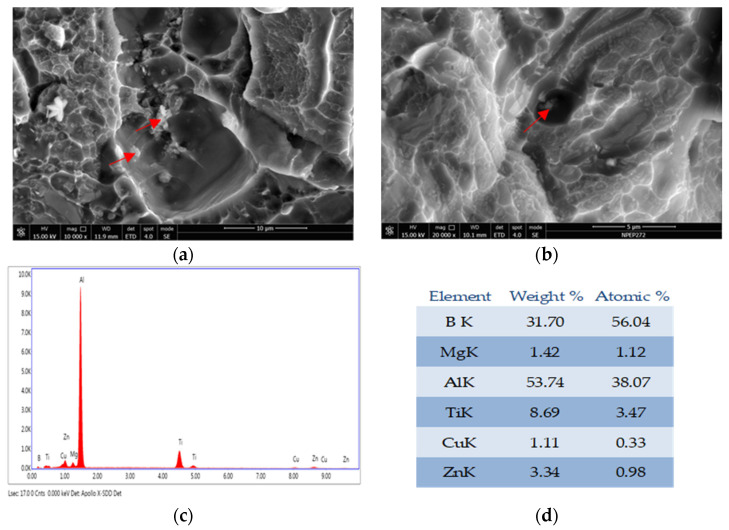
T-L Fracture surface for 7050 plate: (**a**) Al-5Ti-0.2B grain refiner, (**b**) Al-3Ti-0.15C grain refiner (**c**,**d**) EDS result of particles from (**a**), (**e**,**f**) EDS result of particles from (**b**).

**Figure 10 materials-14-06705-f010:**
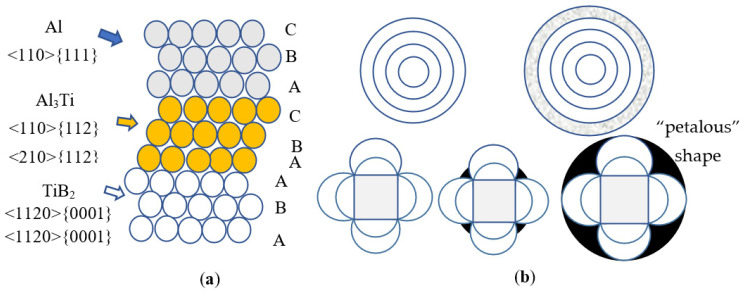
Grain refiner nucleation schematic diagram: (**a**) Orientation relationship between Al, Al_3_Ti, and TiB_2_; (**b**) Nucleation diagram of Al-5Ti-0.2B and Al-3Ti-0.15C refiner during solidification.

**Figure 11 materials-14-06705-f011:**
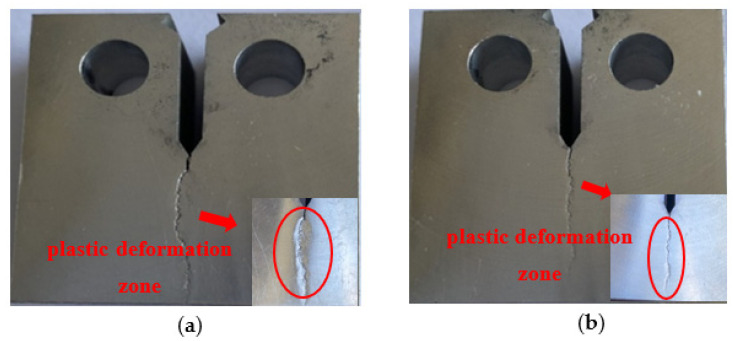
Fracture path for 7050 ingot with different grain refiners: (**a**) Al-3Ti-0.15C (**b**) Al-5Ti-0.2B.

**Figure 12 materials-14-06705-f012:**
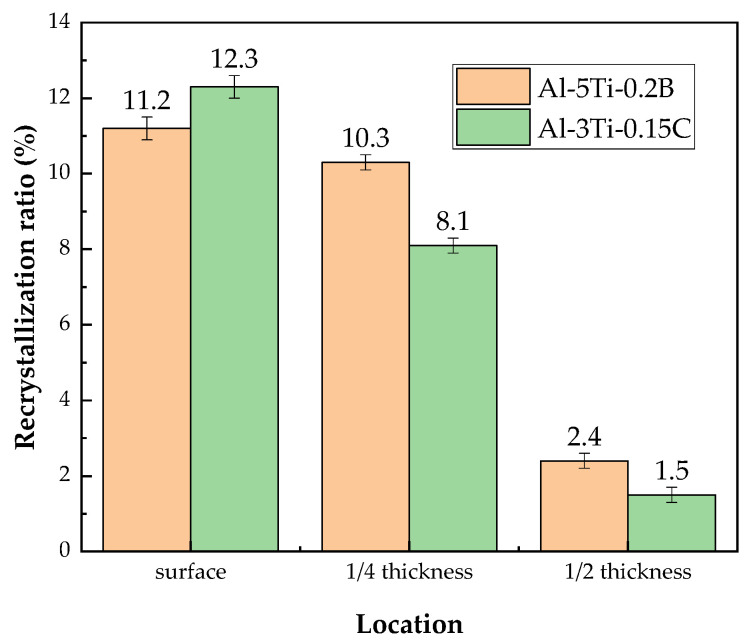
Recrystallization ratio of 7050-T7651 plate with different grain refiners.

**Table 1 materials-14-06705-t001:** Chemical Composition of 7050 Aluminum Alloy (wt%).

Si	Fe	Cu	Mn	Mg	Cr	Zn	Ti	Zr	Al
0.08	0.10	2.2	0.10	2.15	0.02	6.2	0.06	0.10	remainder

**Table 2 materials-14-06705-t002:** Tensile Properties for 7050 Ingot after Treatment with Different Grain Refiners.

Grain Refiner	Ultimate Strength (MPa)	Yield Strength (MPa)	Elongation (%)
Al-5Ti-0.2B	531.0 ± 1.04	474.5 ± 0.50	6.0 ± 0.50
Al-3Ti-0.15C	529.5 ± 0.50	473.6 ± 0.76	7.5 ± 0.50

**Table 3 materials-14-06705-t003:** Tensile Properties for 7050-T7651 100 mm Plate with Different Grain Refiners.

Direction	Al-5Ti-0.2B			Al-3Ti-0.15C		
	Ultimate Strength (MPa)	Yield Strength (MPa)	Elongation %	Ultimate Strength (MPa)	Yield Strength (MPa)	Elongation %
L	532.0 ± 1.44	485.8 ± 2.70	11.3 ± 0.55	522.8 ± 2.44	470.5 ± 2.81	12.8 ± 0.34
LT	543.3 ± 1.91	482.6 ± 3.01	9.6 ± 0.45	529.7 ± 1.37	464.0 ± 2.06	10.5 ± 0.34
ST	531.2 ± 0.47	449.7 ± 0.24	7.6 ± 0.24	523.5 ± 0.63	438.9 ± 1.86	8.8 ± 0.69

## Data Availability

The data presented in this study are available upon request.

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
