# Peer review of "Effect of Grain Refiner on Fracture Toughness of 7050 Ingot and Plate"

_materials, 2021, doi:10.3390/ma14216705_

Round 1
Reviewer 1 Report
The submitted manuscript discusses the effect of grain refiner (Ti-B composition) on fracture toughness of 7050 ingot and plate. The manuscript seems to have dramatic English writing problems as could be seen in the attached file that contains the reviewer's report. The use of terms was not consistent all the way through the manuscript and the experimental procedure is not well organized. Furthermore, many findings in the results part require discussion and explanation. General comments are listed below, and detailed comments could be found in the attached file:
1- The abstract is not clear. It needs extensive English editing. Furthermore, many information are missing. Authors are strongly requested to refine the English language and rewrite the abstract in more concise manner.
2- Why to use different alloy designation??? the title mentions it is 7050 Al alloy, while the text states it is Al-Zn-Mg-Cu alloy.
3- The references are not cited properly in the text. Please refer to the Instructions for authors to learn how references are cited in the text.
4- Insufficient literature data was provided. Furthermore, the literature data is scattered and out of the scope in many locations.
5- Authors must organize the steps of experimental procedure. 6- Lines 108-110: Italic font, is there any reason for that??? 7- Improper caption of Figure 6. Authors are requested to use a concise and self explanatory caption.
Author Response
Dear editor,
Thank you and all reviewers for editing and reviewing our manuscript. The valuable comments really help to increase the quality of the manuscript. We have revised the manuscript based on reviewers’ comments and suggestions and submit the revised manuscript for your consideration for publication.The revisions are highlighted in the revised manuscript and also explained in the attached “Response to Reviewers Comments”. We hope that the revised manuscript is satisfactory for publication. However, we warmly welcome any further requests and suggestions. .
Thanks again for giving us chance to revise our manuscript
Best wishes,
Yu Fang,
Doctor,
School of Material Science and Engineering,
Northeastern University
Contact Number: (+86)16653558810
Email: yufang1989@163.com

Reviewer 2 Report
Dear,
In general, the paper is interesting.
The manuscript must be improved, there are some missing relevant described, and a lot of style mistake (no blank space, not SI system, formatting, upper latter)
I done full review in PDF, so please check it.
Specific comments are given as follows:
48 - Some additional explanation from literature about refination mechanisms of and achieved properties is required.
84 - aging parameters
98 - values are different on figure 4
Missing measurement for non refiner specimen to compare
148 - How detect particles? no EDS result.
238 - There is no reported results of dispersion phases in this paper.
Again, please check attached PDF

Author Response

(The authors gave the same response as above.)

Round 2
Reviewer 1 Report
The authors put lots of efforts to answer the reviewer's comments. The revised manuscript is improved indeed. However, the English is still poor and this is clear from the modifications made in the abstract section. Furthermore, the references are still improperly cited in the text and others are missing. Please see attached file.
With respect to comment number 31 in the reviewer's report round 1, yes please add the reference for AMS 2355 and for the reference in Line 72, in the revised manuscript.

Author Response
Dear Reviewer,
We are submitting the revised manuscript for your re-consideration for publication.The revisions are highlighted with blue color in the revised manuscript and also explained in the attached “Response to Reviewers Comments”. We hope that the revised manuscript is satisfactory for publication. We are sincerely appreciate the hard work and guidance.
Best regards,
Yu Fang,
Doctor,
School of Material Science and Engineering,
Northeastern University
Contact Number: (+86)16653558810
Email: yufang1989@163.com

Reviewer 2 Report
Now the manuscript is improved, but there are some jobs to do, improve figure 2 and correct of style mistake (no blank space, formatting, upper letter)
Please check PDF for comment (cyan colour)

Author Response

(The authors gave the same response as above.)
